# Depression and Pseudodementia: Decoding the Intricate Bonds in an Italian Outpatient Setting

**DOI:** 10.3390/brainsci13081200

**Published:** 2023-08-13

**Authors:** Beatrice Buccianelli, Donatella Marazziti, Alessandro Arone, Stefania Palermo, Marly Simoncini, Manuel Glauco Carbone, Leonardo Massoni, Miriam Violi, Liliana Dell’Osso

**Affiliations:** 1Department of Clinical and Experimental Medicine, Section of Psychiatry, University of Pisa, 56126 Pisa, Italy; bbuccianelli@gmail.com (B.B.); alessandroarone2@gmail.com (A.A.); stefania.palermo@hotmail.com (S.P.); marlysimoncini@virgilio.it (M.S.); lmassoni700@gmail.com (L.M.); miriamvioli@gmail.com (M.V.); liliana.dellosso@unipi.it (L.D.); 2Saint Camillus International University of Health and Medical Sciences—UniCamillus, 00131 Rome, Italy; 3Department of Medicine and Surgery, Division of Psychiatry, University of Insubria, 21100 Varese, Italy; manuelglaucocarbone@gmail.com

**Keywords:** pseudodementia, dementia, cognitive dysfunction, depression

## Abstract

In spite of the uncertainties of its diagnostic framework, pseudodementia may be conceptualized as a condition characterized by depressive symptoms and cognitive impairment in the absence of dementia. Given the controversies on this topic, the aim of the present study was to assess neurological and cognitive dysfunctions in a sample of elderly depressed subjects, and the eventual relationship between cognitive impairment and depressive symptoms. Fifty-seven elderly depressed outpatients of both sexes were included in the study. A series of rating scales were used to assess diagnoses, depressive and cognitive impairment. Comparisons for continuous variables were performed with the independent-sample Student’s *t*-test. Comparisons for categorical variables were conducted by the χ^2^ test (or Fisher’s exact test when appropriate). The correlations between between socio-demographic characteristics and clinical features, as well as between cognitive impairment and depressive symptoms were explored by Pearson’s correlation coefficient or Spearman’s rank correlation. Our data showed the presence of a mild–moderate depression and of a mild cognitive impairment that was only partially related to the severity of depression. These dysfunctions became more evident when analyzing behavioral responses, besides cognitive functions. A high educational qualification seemed to protect against cognitive decline, but not against depression. Single individuals were more prone to cognitive disturbance but were similar to married subjects in terms of the severity of depressive symptoms. Previous depressive episodes had no impact on the severity of depression or cognitive functioning. Although data are needed to draw firm conclusions, our findings strengthen the notion that pseudodementia represents a borderline condition between depression and cognitive decline that should be rapidly identified and adequately treated.

## 1. Introduction

Pseudodementia is a psychopathological condition frequently encountered in clinical practice. It is characterized by the co-presence of depressive symptoms and of cognitive and functional impairment that mimics dementia, but unlike dementia, it may be reversible [1,2,3]. It is still a controversial topic in neuropsychiatry and, as such, subject to continuous revisions and debates within the scientific community.

The major clinical features of pseudodementia include the presence of cognitive symptoms in individuals diagnosed as depressed, or vice versa, coupled with the absence of clearcut evidence of a neurodegenerative disease. Given that dementia shows depressive symptoms, that depression may be characterized by cognitive impairment and that there might be a clinical overlap between the two conditions; their diagnostic distinction is often difficult and represents an unresolved issue. Although there is still an ongoing debate on the boundaries between depression and dementia and on their reciprocal influences, in any case, there is a general agreement on the use of the “pseudodementia” term, as it would suggest, at least to a certain extent, shared neurobiological underpinnings in the two conditions [4].

The prevalence of depressive disorders in elderly people is around 1–13%, up to 15% in hospitalized subjects, and even as high as 35% amongst inpatients of healthcare facilities [5]. Different risk factors for depressive disorders have also been identified in the elderly, including female gender, stressful events (such as bereavement and retirement), and medical illnesses [6,7]. As compared with other age groups, elderly depressed patients may more frequently show somatization symptoms, decreased appetite, asthenia, and irritability during the affective episode [8,9]. Cognitive symptoms are also frequent in the elderly depressed patients that may often show impairment in different executive functions, such as planning, abstraction, and organization [10,11,12,13,14,15,16]. To complicate the matter, it should be highlighted that the onset of depressive symptoms in the elderly may also represent the prodrome of that type of dementia called “vascular” which is frequently associated with vascular damage [17] and with evident alterations of the white matter, as shown by magnetic resonance imaging (MRI) studies [18,19]. Not surprisingly, depression is considered to represent a risk factor not only for vascular, but also for Alzheimer’s dementia (AD) [20,21,22,23].

Depression is quite common in patients with different types of dementia and more prevalent than in age-matched healthy controls. The incidence is about 30% in both vascular dementia and AD, and higher (up to 40%) in Parkison’s disease [24,25,26].

From the neurobiological point of view, some inconclusive data support the notion that there might exist common etiological and pathophysiologic mechanisms in depression and dementia, encompassing monoamine deficits, inflammatory and neurotrophic processes, and vascular damage [27,28,29,30,31,32,33,34]. What is currently known is that depression might promote cognitive decline by inducing hippocampal and frontal atrophy via the glutamatergic system, albeit with no relationship with plaques or tangle alterations [35,36,37,38]. The involvement of some common etiological factors and reciprocal influences between depression and dementia is also plausible, in line with the so-called “cognitive reserve theory”. Cognitive reserve is a term used to denote individual differences in performing tasks making some subjects less vulnerable to age-related brain changes or dementia. According to this notion, brain aging might cause a reduced ability to compensate for impaired cognitive symptoms that might become pathologic in vulnerable individuals. Depression might be one of the factors inducing this kind of increased susceptibility to dementia [25,39].

Nevertheless, drawing a line between the two conditions is not an easy task. According to Kaszniak [40], the difficulty may be due to the frequent presence of a cognitive decline in the elderly, or to the evidence that pseudodementia may be confused with normal brain aging, or to the frequent occurrence of psychopathological symptoms in several neurodegenerative diseases [41,42]. Some features in the clinical history may assist to formulating a correct diagnosis, such as a family or personal history of mental disorders, a good response to psychopharmacological drugs, a greater tendency to complain of hyporexia, marked mental fatigue, and a similar alteration of both anterograde and retrograde memory, unlike, for example, AD patients [43,44,45]. In addition, standardized methods are increasingly being used to ease the differential diagnosis. These include neuropsychological testing indicating that patients with pseudodementia show less impairment in processing speed, memorymemory, and attention than patients with dementia. In addition, neuroimaging techniques demonstrate that demented patients may have a diffuse rather than a focal reduction in blood flow. Other differences have also been recorded in REM sleep characteristics, electroencephalography, and event-related potentials to unexpected stimuli [46,47,48,49].

A review of longitudinal studies showed that 62% of patients with pseudodementia experienced an improvement or stability in their psychiatric picture over time, while 38% of patients developed an irreversible form of dementia [50]. It was then concluded that pseudodementia might confer a certain risk for a progression towards dementia, although this is not unavoidable and requires the need to promptly distinguish between the two conditions [50]. As such, a correct differential diagnosis is crucial, as therapeutic strategies for pseudodementia do exist, i.e., antidepressant drugs that may successfully cause a remission of the depressive symptomatology [50]. Specifically, selective serotonin reuptake inhibitors (SSRIs), serotonin and noradrenaline reuptake inhibitors (SNRIs), and vortioxetine have has been demonstrated to be effective [16,51,52,53,54,55,56,57,58].

However, some warnings have been published on the possible risks that some antidepressants, in particular tricyclics, might increase the risk of dementia beyond depression itself [59,60].

Given the current controversies and unanswered questions concerning pseudodementia, the present research aimed to evaluate the presence of psychopathologic symptoms and cognitive impairment by means of specific scales and questionnaires in a group of elderly depressed outpatients. The possible relationships between socio-demographic characteristics and clinical features, as well as between cognitive impairment and depressive symptoms, were also explored to possibly suggest tailored therapeutic interventions that might improve both conditions.

## 2. Materials and Methods

The present study included a sample of elderly depressed patients recruited in the outpatient ward of Pisa University Hospital who requested a psychiatric consultation for a major depressive episode. The patients were evaluated with different rating scales to assess both psychiatric symptoms and cognitive functions.

### 2.1. Patients

The study included 57 patients older than 65 years who consulted our psychiatric outpatient unit from December 2020 to June 2022 for heterogeneous first-onset depressive symptoms, or for a cross-sectional episode in the context of a longitudinal diagnosis of mood disorders. All eligible subjects were first assessed through a one-hour-long clinical interview and received a diagnosis of a major depressive episode (MDE) according to the Diagnostic and Statistical Manual of Mental Disorders fifth edition (DSM-5) criteria [61]. The inclusion criteria were the following: age over 65 years, patients with a diagnosis of a major mood episode according to DSM-5 criteria, patients with neurological symptoms that could not be framed with a specific diagnosis, patients with neurological symptoms that could be framed with a specific diagnosis, patients able to sign an informed consent. The exclusion criteria were: age under 65 years, patients unable to sign an informed consent, patients with alcohol or other substance abuse in the last 12 months. Exit criteria were as follows: withdrawal of informed consent and evidence of exclusion criteria during investigations.

Medical, psychiatric, pharmacological, family and personal histories (gender, age, level of education, marital status) were also collected as an integral part of the psychiatric assessment.

After a complete description of the study, written informed consent was obtained from each patient to participate in the study previously approved by the Ethics Committee at Pisa University.

### 2.2. Assessment Scales

The following rating scales were used:

- *Mini-International Neuropsychiatric Interview* (MINI) [62]: It is an easy -to- use and rapid screening tool tailored to explore major psychiatric symptoms and to diagnose several psychiatric disorders. It is divided into several modules, all including one or two preliminary screening questions, and followed by questions aimed at detecting specific symptoms and then assessing an impairment in functioning and concomitant pathologies or substance abuse.

- *Hamilton Depression Rating Scale* (HAM-D or HRSD) [63]: This 21-items questionnaire is used worldwide to assess the severity of depression. The severity cut-off is structured as follows: ≥25 severe depression; 18–24 moderate depression; 8–17 mild depression; ≤7 no depression.

- *Beck Inventory Scale* (BDI) [64]: The scale was specifically designed to measure the behavioral manifestations of depression in patients fulfilling the clinical diagnostic criteria for depressive syndromes. The cut-off scores for BDI are: scores of 0 to 9 indicate absence of or minimal depression; scores of 10 to 18 indicate mild to moderate depression; scores of 19 to 29 indicate moderate to severe depression; and scores of 30 to 63 indicate severe depression.

- *Geriatric Depression Scale* (GDS) [65]: It is a self-administered scale designed to assess depression in the elderly. A total score of up to 10 indicates the absence of depression; scores between 11 and 13 indicate the possible presence of depression; with scores of 14 or more, the presence of depression is certain.

- *Short Psychiatric Evaluation Scale* (SPES) [66,67]: This scale is a complement to the Short Portable Mental Status Questionnaire (SPMSQ), designed to assess organic mental deficits in older people, given that they often show functional mental symptoms. A total score of up to 4 is equivalent to the absence of a clinically relevant psychopathology; scores of 6 or more indicate the certain presence of psychopathology; a score of 5 does not allow the exclusion of psychopathology, but neither denies its presence.

- *Cornell Scale for Depression in Dementia* (CSDD) [68]: It aims to assess depression in people with dementia, and it is based on direct observations and interviews with the patient or a reliable informant or clinician. The CSDD is made of 19 items that refer to the five cores of depression (mood symptoms, behavioral disorders, somatic symptoms, circadian rhythms and cognitive symptoms). The items are rated on a 3-level scale, from 0 = absent to 2 = severe, so the total score can vary from 0 to 38. The higher the score, the greater the severity of depression, while a score of 12 indicates moderate depression and a score of 8 suggests a mild severity.

- *Montreal Cognitive Assessment* (MoCA) [69]: It is a cognitive screening tool tailored to assess mild cognitive impairment (MCI). It includes 30 questions that explore different cognitive skills: orientation, short-term memory/delayed recall, executive functions/visuospatial skills, language skills, abstraction, fluidity, and attention. The total score is 30, with a score greater than 26 being considered normal. It is a first-level battery of tests tailored to examine the global operational functioning, and is composed of both cognitive and behavioral tests. It is also a useful tool in discriminating fronto-temporal dementia from AD in subjects with mild dementia. Each test has a score from 0 to 3, with a maximum total of 18 points.

- *Frontal Assessment Battery* (FAB) [70]: It is a first-level battery of tests tailored to examine global operational functioning and is composed of both cognitive and behavioral tests. It is also a useful tool for discriminating between fronto-temporal dementia from AD in subjects with mild dementia. Each test has a score from 0 to 3, with a maximum total of 18 points. The higher the score, the less executive functions are impaired.

- *Mini-Mental State Examination* (MMSE) [71]: It is a valid and reliable test to explore cognitive functions that are also sensitive to changes in time. Despite being validated and used mostly in subjects with organic mental problems, MMSE has also been shown to be flexible in the evaluation of cognitive functions in subjects suffering from mood disorders or schizophrenic spectrum disorders. It consists of 11 items divided into two parts (verbal and performance). Each item has a score ranging from 0 to 1 or 0 to 5, with a maximum score of 21 in the first part and 9 in the second. The threshold score for “normality” is set at 24/30; however, this limit is influenced by age and schooling, for which correction factors have been developed based on age and the individual level of schooling.

### 2.3. Statistical Analyses

All demographic and clinical data were presented for continuous variables in terms of mean ± standard deviation (SD), variation range (min and max values), or medians, when required. Categorical variables were expressed as frequencies (numbers) and percentages.

The Kolmogorov-SmirnovKolmogorov–Smirnov test was used to determine the normality of the distribution of the variables. Comparisons for continuous variables were performed with the independent-sample Student’s *t*-test. Comparisons for categorical variables were conducted by the χ^2^ test (or Fisher’s exact test when appropriate).

The correlations between the different features of the subjects and psychopathological dimensions were explored by calculating Pearson’s correlation coefficient or Spearman’s rank correlation. Pearson’s correlation is used to measure the degree of the relationship between linearly related variables. Spearman’s rank correlation is a non-parametric test that is used to measure the degree of association between two variables. Spearman’s rank correlation test does not carry any assumptions about the distribution of the data and is the appropriate correlation analysis when the variables are measured on a scale that is at least ordinal. The assumptions of the Spearman’s Spearman’s correlation are that data must be at least ordinal and the scores on one variable must be monotonically related to the other variable. Cohen’s d standard may be used to evaluate the correlation coefficient to determine the strength of the relationship, or the effect size. Correlation coefficients between 0.10 and 0.29 represent a small association, coefficients between 0.30 and 0.49 represent a medium association, and coefficients of 0.50 and above represent a large association or relationship. All statistical analyses were carried out using SPSS, version 27 (IBM Corp. Released 2020. IBM SPSS Statistics for Windows, Version 27.0. Armonk, NY, USA: IBM Corp.).

## 3. Results

### 3.1. Clinical Characteristics of the Patients

Our sample included 57 outpatients, of whom 37 were men and 20 were women (mean age + SD: 74.98 ± 5.27 years).Thirty). Thirty-five (61.4%) patients were married, fifteen (26.3%) were separated or divorced, four (7%) were widowed, and three single (5.3%).

Thirty-eight (66.7%) had a familiar history of mental disorders and 19 nineteen (33.3%) had none. Twenty-four (42.1%) patients were suffering from bipolar disorders (BDs) and 33 thirty-three (57.9%) from major depressive disorder (MDD). Twenty-five (43.9%) patients were at their first major depressive episode, while the remaining 32 thirty-two (56.1%) had a history of mood disorders (MDs). Fifty (87.7%) patients had no psychiatric comorbidity, while seven (12.3%) had at least one comorbidity, specifically four (7.0%) were diagnosed with panic disorder, and three (5.3%) with generalized anxiety disorder. Thirty-five (61.4%) reported at least one medical comorbidity, and 22 twenty-two patients (38.6%) had no medical disease. Cardiovascular disease was present in 27 (47.4%) and diabetes mellitus in four 4 patients (7.0%). Five patients (8.8%) were also suffering from a neurological disease (stroke, essential tremor, Parkinson’s disease, chorea) (Table 1).

At the time of the observation, the patients were treated with different psychotropic drugs, according to the current episode and the possible longitudinal mental disorder. Thirty-five patients (61.4%) were treated with a mood stabilizer (valproic acid, lithium salts, gabapentin), and six of these (10.5%) with two mood-stabilizers. Forty-nine patients (86%) were treated with antidepressants (SSRIs, SNRIs, tricyclics, mirtazapine), 22 twenty-two (38.6%) with an antipsychotic (quetiapine, clozapine, perphenazine, olanzapine, risperidone), 14 fourteen with benzodiazepines (24.6%), and one patient (1.8%) with an acetylcholinesterase inhibitor (AchEI) (Figure 1).

Only three patients (5.3%) underwent neuroradiological investigations that, however, did not reveal any organic alterations.

### 3.2. Psychopathological and Neurocognitive Features

The HAM-D total score (mean ± SD) was 12.18 ± 6.33, the BDI total score (mean ± SD) was 12.79 ± 9.89, the GDS total score (mean ± SD) was 12.69 ± 8.25, and the CSDD total score (mean ± SD) was 8.35 ± 6.25. These values indicate the presence of a mild severity of the depressivedepressive symptomatology. The total score of the SPES, assessing the mental state of elderly subjects with organic mental deficits, resulted to be 5.75 ± 3.82 (mean ± SD): this value might be considered inconclusive both for excluding or confirming the presence of a psychopathological condition, as the normal range is 4–6 (Table 2).

The psychometric evaluation of the cognitive functions was performed by means of MoCA, FAB, and MMSE. The MoCA total score (mean ± SD) was 21.30 ± 4.86, that ofthe FAB score was 14.12 ± 3.92, and that ofthe MMSE score was 25.06 ± 4.20. Only tThe MoCA scores were lower than the normal range (>26) (Table 2 and Figure 2).

### 3.3. Correlations between Depressive and Cognitive Symptoms

The HAM-D total score was positively related to that of BDI (r = 0.77: *p* < 0.01), GDS (r = 0.68: *p* < 0.01) and CSDD (r = 0.70: *p* < 0.01). Similarly, the BDI total score was positively correlated with the GDS (r = 0.77: *p* < 0.01) and CSDD (r = 0.73: *p* < 0.01) scores. Furthermore, the GDS total score also positively correlated with the CDSS total score (r = 0.67: *p* < 0.01) (Table 3, panel a).

The MoCA total score was positively correlated with the FAB (r = 0.75: *p* < 0.01) and with the MMSE (r = 0.52: *p* < 0.01) total scores. Moreover, the FAB total score was positively related to the MMSE total score (r = 0.33: *p* < 0.01) (Table 3, panel b). No other correlations were detected.

The evaluation of the possible correlations between the scales assessing depressive symptoms and those evaluating the neuropsychological profile showed a positive correlation between the FAB and the HAM-D (r = 0.29: *p* < 0.01), BDI (r = 0.26: *p* < 0.01) and CSDD (r = 0.34: *p* < 0.01) (Table 3, panel c).

By using the Mann-WhitneyMann–Whitney test for the analysis of quantitative variables in independent samples, after dividing the sample into single (widowed, separated or unmarried) and married patients, it was noted that married individuals showed higher scores on the MoCA test (Z = 2.30; *p* = 0.021). Patients with a high school diploma or college/university degree, when compared to the rest of the sample, showed higher scores on the HAM-D (Z = 2.21; *p* = 0.027), SPES (Z = 2.11; *p* = 0.034), FAB (Z = 4.54; *p* < 0.01) and MoCA tests (Z = 3.56; *p* < 0.01)

No significant differences emerged regarding the presence/absence of familiar history for mental disorders, classes of psychotropic drugs, medical comorbidities, number of affective lifetime episodes, and gender.

Taken together, our findings, partly in agreement with the available literature, indicate that a mild cognitive impairment is common in elderly depressed patients, although unrelated to the affective symptoms.

## 4. Discussion

The present research study aimed at assessing the presence of cognitive impairment in a sample of 57 elderly outpatients of both sexes who were consulting the psychiatric unit of an Italian university hospital for a major depressive episode. The possible correlation between cognitive impairment and the severity of the depressive episode was also investigated. This study is valuable even because, to our knowledge, the information on this topic in our country is limited to case reports, reviews, or Delphi studies [55,72,73,74].

The comorbidity of depressive symptoms with cognitive impairment is a common condition in the clinical practice that is called “pseudodementia” to highlight that, unlike “true” dementia, it may be reversible [1,2,3]. Pseudodementia is a controversial topic, because, in spite of its definition implying the absence of any evident sign of neurodegeneration, it has also been interpreted as an early sign of dementia [4,17,18,19,20,21,22,23]. Interestingly, to disentangle part of the question, European guidelines recommend using brain FDG-PET to differentiate between AD and depression [75].

Our sample included 37 men and 20 women of about 75 + 5 years of age, who were mainly married (n. 35) and had a family history of psychiatric disorders (n. 38), with no differences between the two sexes. Thirty-three patients were suffering from BDs and twenty-four from major depressive disorder. The majority (n. 32) of our patients had a previous history of MDs, while 25 were at their first major depressive episode. No difference was noted in both depressive and cognitive symptoms between first-episode patients and patients with multiple mood episodes. These findings are in contrast with a previous study reporting a greater severity of subsequent episodes when compared to the first [76]. The patients were treated with psychotropic drugs or different combinations of them, according to clinical needs. Three patients had already undergone neuroradiological investigations that did not reveal any neurodegenerative disease.

As far as psychopathological symptoms are concerned, our findings showed that scores of the scales assessing depression (HAM-D, BDI, GDS, CSDD) were just above the diagnostic levels, thus indicating that our patients were suffering from mild–moderate depression. A possible explanation could be that, when filling self-administered tests, the patients might underestimate and consider some symptoms ”normal” (such as vegetative ones) and/or some aspects of their lives (such as adjustment functioning) that the clinicians rate as pathological. In addition, it can be assumed that our patients requested a psychiatric evaluation following advice from one of their family members or general physicians, or during the assessment of a polymorphic picture consisting of, but not only, depressive symptoms, with the psychiatric consultation often representing the last step of a long and distressing diagnostic process. The mean score of the SPES was 5.75 ± 3.82 which should not be considered conclusive to either exclude or confirm the presence of psychopathology. Indeed, it should be underlined that the SPES is a specific test evaluating the mental state of elderly people with organic deficits; therefore, it might result as inadequate to detect and assess the psychopathological symptoms of our patients who displayed some mild cognitive impairment, but no definite organic underpinnings.

The psychometric evaluation of the cognitive functions was performed by means of a battery of standardized psychometric rating scales including MoCA, FAB, and MMSE. The results showed that the scores of the FAB and MMSE were within the normal range, while pathological scores were only noted on the MoCA, with a mean score of 21.30 ± 4.86 (normal range > 26). This result would indicate the presence of cognitivedifficulties when assessing spatial–temporal orientation, attention, concentration, working memory, memory recall tasks, visual–spatial skills, language, and executive functions. It is also true that MoCA is considered one of the most sensitive screening tests to detect mild cognitive impairment (MCI). Our findings can be considered in agreement with the current literature—and our hypothesis—that cognitive functions, particularly executive functions, are frequently impaired during a depressive episode [77]. It is not surprising that the MMSE total score was 25.06 ± 4.20, only slightly above the threshold of 24, as this test is especially used to exclude AD. Therefore, as such, it is not totally reliable for the sharp assessment of those executive functions that are mostly, albeit mildly affected during depression. The mean score of the FAB was 14.12 ± 3.92 (pathological value < 12.03). In this case, we did expect scores suggesting some alterations in elderly depressed patients, as the FAB is a first-level battery of tests to assess the global operational functioning, but our hypothesis was not fulfilled by the actual finding.

By analyzing the correlations between the different psychometric scales, all the scales assessing depressive symptoms were interrelated, as were the scales assessing cognitive symptoms. The analysis of the intergroup relationships revealed that only the FAB (whose score was normal) showed a statistically significant correlation with the HAM-D, BDI, and CSDD total scores, while no correlations were found between the HAM-D, BDI, and CSDD total and the MoCA or MMSE scores. These data did not confirm the hypothesis implying linearity between depression and cognitive impairment. It can be supposed that a mild depression, as in our sample, might be linked to alterations in executive functions, as indicated by the MoCA scores, although these dysfunctions are not strong enough to worsen the clinical picture of depression. It can also be hypothesized that the behavioral changes, as assessed by the FAB, might have a greater impact on the correlation between depression and cognitive impairment. Another alternative explanation might be that they have an earlier onset than the cognitive changes, or that they are more easily detectable.

When we analyzed our sample of patients while using the Mann-WhitneyMann–Whitney test for the analysis of quantitative variables in independent samples, some intriguing results emerged. After distinguishing whether the patients were married or single (a wide category including widowed, separated, or unmarried), interestingly, no difference in depressive symptoms emerged, a finding that is totally opposite to that widely reported in the literature [78]. This might be due to the strong family ties typical of our country that may replace the role of a partner for single individuals [79]. However, our patients without a partner showed statistically lower scores on the MoCA, when compared to married ones; this finding confirms the literature that being single or divorced has a negative impact on cognitive functions [80]. When taking into account educational levels, patients with more years of study showed higher scores on both depression (HAM-D) and cognitive scales (MoCA, FAB). Our findings are not in agreement with the reports that a high level of education might protect against depression [81,82,83], while they are congruent with the evidence that a long study period is a positive factor against brain aging [81]. It can be hypothesized that, as shown by surveys and preventive interventions against mental illness amongst university personnel, patients with more years of education show a higher tendency to reflect, to ruminate, to doubt, to meditate around existential issues, and a lower tolerance to psychological discomfort that may easily turn into depression (or other psychopathological disturbances) in predisposed individuals [84,85].

There are some limitations of this study that should be acknowledged, such as the small sample size, the use of largely self-administered tests, and the scanty use of instrumental investigations that might detect early cognitive aging. However, it should be underlined that this is one of the few types of research carried out in our country on the important topic of pseudodementia. Moreover, almost half of the subjects were suffering from BDs; this entails the presence of possible cognitive alterations caused by previous manic episodes that were not analyzed, since we focused on the current depressive episode. Last, given the small sample size, we could not analyze the possible impact of different psychotropic drugs or drug combinations on cognitive functions, which is another controversial topic in neuropsychiatry [86,87]

To sum up, our sample of elderly Italian outpatients suffering from moderate depression with a major diagnosis of major depressive disorders and BDs, mainly treated with mood stabilizers and antidepressants, showed mild alterations in cognitive functions that resulted only partially related to the severity of the depressive episode. The cognitive impairment was mild and only evident in the MoCa total score. These mild dysfunctions became more evident when the behavioral responses were analyzed by the FAB, a specific scale including a battery of tests tailored to examine global operational functioning. According to our findings, we would suggest that the MoCa and the FAB seem to be the most appropriate and sensitive rating scales to detect MCI early, typical of elderly depression.

In our sample, being single was not a risk factor for depression, but for cognitive impairment. High educational qualifications were related to more severe depressive symptoms.

Our data seem to suggest that previous episodes do not necessarily imply a more severe depression or worse cognitive functioning. This might be explained by the protective and neurotrophic functions exerted by some, but not all antidepressants and antipsychotics against brain aging [28,59,60,88]. However, their role in preventing the onset of dementia remains controversial. Although the early treatment of depressive symptoms seems to be useful not only for depression itself, but for the quality of life of the patients [32,33,34,41,52,53,55,56], recent studies raise the question of whether antidepressants (at least those with anticholinergic activity) increase the risk of dementia or not in elderly depressed subjects [28,59,60].

## 5. Conclusions

Pseudodementia is still a controversial topic far from being fully clarified [3,16]. The overall findings of our study would indicate that it constitutes a borderline condition between depression and cognitive decline that can be found in several geriatric patients attending psychiatric outpatient units. Although disagreement does exist [28,31,56,58,59,60], our suggestion is that it seems pivotal to promptly diagnose and detect early depressive symptoms in elderly patients with or without cognitive impairment, and to treat them correctly. Indeed, if it is true that depressed and older patients with dementia, especially men, are at an increased risk of developing dementia [27,28], and that depressive symptoms in the elderly may be prone to neurodegenerative disorders [30,89]; we cannot avoid targeting a modifiable risk factor [17,32,33,38,57]. In any case, a multidisciplinary approach involving psychiatrists, neurologists, neuroradiologists, and geriatricians is necessary to better take care of patients (and relatives) [34,41,42] in a treatment trajectory that does not follow pre-established rigid guidelines but is based on clinical evidence. To reach an exhaustive approach to the problem of pseudodementia, more accurate and sensitive assessment tools are needed to enable clinicians to rapidly detect those clinically subtle changes that might be the first signs of MCI. The prompt availability of neuroimaging techniques is also essential in controversial cases. It should also be of paramount importance to gather more preclinical and clinical data on the “real” neurotrophic (or detrimental) potential of different classes of psychotropic drugs. Overall, these factors should promote a tailored therapeutic approach to individual cases of pseudodementia.

Currently, we are following up the patients included in the present study, as well as other ones, to regularly assess cognitive functions and depressive symptoms, together with a careful monitoring of drug treatment, to explore the outcomes over time.

## Figures and Tables

**Figure 1 brainsci-13-01200-f001:**
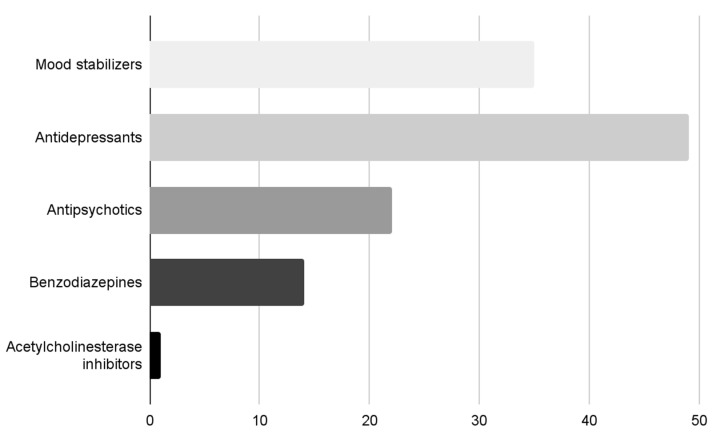
Classes of psychotropic drugs of the patients.

**Figure 2 brainsci-13-01200-f002:**
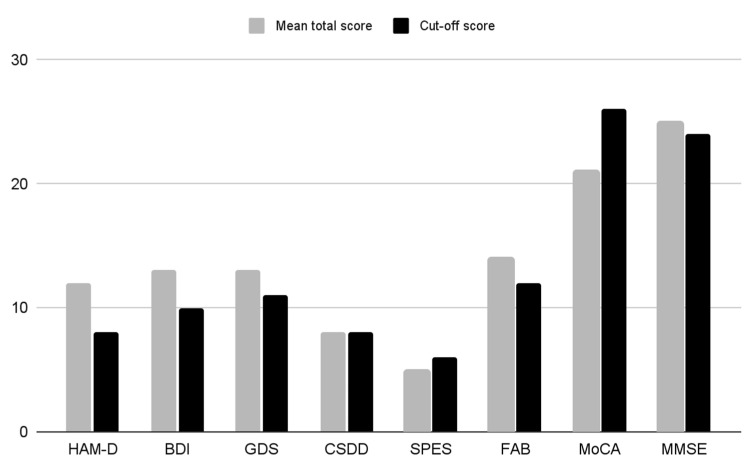
Histogram of the mean total scores of the rating scales. HAM-D (Hamilton Rating Scale for Depression; BDI (Beck Inventory Scale); GDS (Geriatric Depression Scale); CSDD (Cornell Scale for Depression in Dementia); SPES (Short Psychiatric Evaluation Scale); FAB (Frontal Assessment Battery); MoCA (Montreal Cognitive Assessment); MMSE (Mini-Mental State Examination).

**Table 1 brainsci-13-01200-t001:** Demographic characteristics of 57 elderly depressed patients.

N (%)		
Diagnosis	Depressive disorder	33 (57.9%)
	Bipolar disorder	24 (42.1%)
Number of episodes	First episode	25 (43.9%)
	Multiple episodes	32 (56.1%)
Family history of psychiatric disorder	None	38 (66.7%)
	Positive	19 (33.3%)
Psychiatric comorbidity	None	50 (87.7%)
	Panic disorder	4 (7%)
	Generalized anxiety disorder	3 (5.3%)
Neurological disease	None	52 (91.2%)
	Stroke, chorea, Parkinson’s disease, essential tremor	5 (8.8%)
Medical illness	None	22 (38.6%)
	One medical disease	35 (61.4%)

**Table 2 brainsci-13-01200-t002:** Clinical and neurocognitive assessments of 57 elderly depressed patients.

Scale	Mean Total Score ± SD	Cut-Off Score
HAM-D	12.18 ± 6.33	≥8
BDI	12.79 ± 9.89	≥10
GDS	12.69 ± 8.25	≥11
CSDD	8.35 ± 6.25	≥8
SPES	5.75 ± 3.82	≥6
FAB	14.12 ± 3.92	<12.03
MoCA	21.30 ± 4.86	<26
MMSE	25.06 ± 4.20	≤24

**Table 3 brainsci-13-01200-t003:** Correlation between rating scales assessing depressive symptoms (panel a), cognitive (panel b) symptoms and depressive and cognitive symptoms (panel c) (only significant data are reported).

**Panel a**									
	**BDI**			**GDS**			**CSDD**		
	**r**	** *p* **		**r**		** *p* **	**r**		** *p* **
HAM-D	**0.77**	**<0.01**		**0.68**		**<0.01**	**0.70**		**<0.01**
BDI				**0.77**		**<0.01**	**0.73**		**<0.01**
GDS							**0.67**		**<0.01**
**Panel b**									
	**FAB**				**MMSE**				
	**r**		** *p* **		**r**			** *p* **	
MoCA	**0.75**		**<0.01**		**0.52**			**<0.01**	
FAB					**0.33**			**<0.01**	
**Panel c**									
	**FAB**			**MoCA**			**MMSE**		
	**r**	** *p* **		**r**		** *p* **	**r**		** *p* **
HAM-D	**0.29**	**<0.01**		**ns**		ns	ns		ns
BDI	**0.26**	**<0.01**		**ns**		ns	ns		ns
CSDD	**0.34**	**<0.01**		**ns**		ns	ns		ns

HAM-D (Hamilton Rating Scale for Depression); BDI (Beck Inventory Scale); CSDD (Cornell Scale for Depression in Dementia); FAB (Frontal Assessment Battery); MoCA (Montreal Cognitive Assessment); MMSE (Mini-Mental State Examination); ns (not significant).

## Data Availability

Data generated or analyzed during this study are included in this published article.

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
