# Peer review of "Depression and Pseudodementia: Decoding the Intricate Bonds in an Italian Outpatient Setting"

_brainsci, 2023, doi:10.3390/brainsci13081200_

Round 1
Reviewer 1 Report
Dear author,
Thank you very much for the submission. This review clearly defined the correlation of pseudodementia between depression and cognitive decline which might provide as a pre-diagnosis for neurodegenerative disorders.
Author Response
No major revision required.
We thank very much the reviewer 1 for the positive comments
Reviewer 2 Report
Thanks for recommending me as a reviewer. In this paper, authors were aimed to assess neurological and cognitive dysfunctions in a sample of elderly depressed subjects and possibly the bidirectional relationship between the two conditions. Fifty-seven depressed patients over 65 years of age were recruited in a psychiatric outpatients ward. Specific scales were used to assess both depressive symptoms and cognitive impairment. Appropriate statistical analyses were used for sample description comparisons and correlations assessments. If the authors complete minor revisions, the quality of the study will be further improved.
1. The introduction section is well written. However, some paragraphs are too verbose. If authors separate some paragraphs, they can help readers understand.
2. line 100-110: It is recommended that only the selection and inclusion criteria of the subjects be presented in the Methods section, while the general characteristics of the subjects are more specific in the Results section
It is recommended that only the selection and inclusion criteria of the subjects be presented in the Methods section, while the general characteristics of the subjects are more specific in the Results section3. In Table 2, authors should present the cutoff score for each test.
Author Response
We thank very much the reviewer 2 for the positive comments and minor reviisons requested
The introduction section is well written. However, some paragraphs are too verbose. If authors separate some paragraphs, they can help readers understand. We separated some of the paragraphs to make the reading clearer, as suggested.
line 100-110: It is recommended that only the selection and inclusion criteria of the subjects be presented in the Methods section, while the general characteristics of the subjects are more specific in the Results section. The incluson and exclusion criteria are indicted in the text as follows: The inclusion criteria were the following: age over 65 years, patients with a diagnosis of a major mood episode according to DSM-5 criteria, patients with neurological symptoms that could not be framed with a specific diagnosis, patients with neurological symptoms that could be framed with a specific diagnosis, patients able to sign an informed consent. The exclusion criteria were: age under 65 years, patients unable to sign an informed consent, patients with alcohol or other substance abuse in the last 12 months. Exit criteria were as follows: withdrawal of informed consent, evidence of exclusion criteria during investigations.
We moved the general characteristics of the subjects to the Results section.
In Table 2, authors should present the cutoff score for each test. The cut-offs of the different scales used were inserted in the table.
Reviewer 3 Report
Title: with 57 pts this is best described as pilot
Abstract: need to be completely rewritten for information currently the abstract is information with no real summary of findings inclusive of numerical results. The authors encourage to refrain from judgmental descriptions e.g. “Appropriate statistical analyses” finally some sentences suggested the use of AI so please confirm that ethical use was applicable. Give an informative and fair account of what was done and what was found in the abstract.
Intro: some major literature is missing and to my limited knowledge https://pubmed.ncbi.nlm.nih.gov/31190831/ and https://pubmed.ncbi.nlm.nih.gov/34130356/ and https://pubmed.ncbi.nlm.nih.gov/32944058/ can be used. Describe the scientific context and justification for the reported study. Describe your precise goals and any established hypothesis. Why Italy? Pleas use GBD data on dem/depr.
The topic is not new I don’t see why this study is important.
Line 56 “Cognitive symptoms are also frequent [12],” this need to be fully expended into a paragraph and provide %.
Line 91-92 https://www.ncbi.nlm.nih.gov/pmc/articles/PMC7063578/ pls track work of Nili Solomonov
Intro failed to justify the need of the present study, aim is not clear, and no hypothesis.
Method
Early in the text, highlight the important study design components.
Describe the context, the places, the pertinent times, such as the recruiting, exposure, follow-up, and data collection times. All patients e.g. taken from pisan hospital. Who did diagnosis. Who collected information. Ethical issues with consent? Give the qualifying requirements, as well as the sources and procedures used to choose the participants. Are people with comorbidities excluded e.g. anxiety, ocd, ed.
All outcomes, exposures, predictors, potential confounders, and effect modifiers should be precisely defined. Describe the diagnostic standards, if any.
Describe how the study size was determined.
Describe the methods used to handle quantitative variables in the analyses. Describe the classifications that were chosen, if relevant, and why.
Describe all statistical techniques, including confounding correction techniques.
Describe any techniques used to study interactions and subgroups.
Describe how missing data were handled.
Summarize the main findings in relation to the study's goals.
Describe the study's limitations while considering any possible bias or imprecision sources. Discuss any potential bias's magnitude and direction.
Give a cautious overall interpretation of the findings while considering the goals, restrictions, variety of analyses, outcomes from related studies, and other pertinent data.
Discuss how broadly (externally valid) the study's findings can be applied.
2.2 Assessment scales pls provide better description of each tool and alpha for Italian scales
2.3 Statistical analyses is data normal? If Yes, Why both mean and median used?
Results essential tremor need to be referred to as benign essential tremor
Family history of psychiatric disorder? Who is family?
Table 3 pls convert to correlogram to avoid use of confusing panels
Lines 371+ is unnecessary.
Add doi to references.
NOTE: non of the refs suggested are my own.
Author Response
Reviewer 3
Abstract: need to be completely rewritten for information currently the abstract is information with no real summary of findings inclusive of numerical results. The authors encourage to refrain from judgmental descriptions e.g. “Appropriate statistical analyses” finally some sentences suggested the use of AI so please confirm that ethical use was applicable. Give an informative and fair account of what was done and what was found in the abstract. We agree with the reviewer that the abstract was not informative enough and we modified it.
Introduction
We thank the reviewer for his or her suggestions about he text and teh new references and carefully revised the Introduction.
some major literature is missing and to my limited knowledge https://pubmed.ncbi.nlm.nih.gov/31190831/ and https://pubmed.ncbi.nlm.nih.gov/34130356/ and https://pubmed.ncbi.nlm.nih.gov/32944058/ can be use. The three suggested references were included in the text and in the reference list
Describe the scientific context and justification for the reported study. These points were better explained
Describe your precise goals and any established hypothesis. In the present version of the paper, the goal and hypothesis were clearly reported
Why Italy? Pleas use GBD data on dem/depr. Italy was the country of the authors and of the patients recruited where the study was carried out. The Italian data on the prevalence of depession and dementia are similar to those of several other countries: according to us it is useless to indicate them
The topic is not new I don’t see why this study is important. It is true that the topic is not new, but it remains controversial and deserving to be deepened in clinical studies as ours.
Line 56 “Cognitive symptoms are also frequent [12],” this need to be fully expended into a paragraph and provide %..As mentioned above, the introduction was deeply revised
Line 91-92 https://www.ncbi.nlm.nih.gov/pmc/articles/PMC7063578/ pls track work of Nili Solomonov. It is an intriguing paper that ws included in teh text and in the reference list
Intro failed to justify the need of the present study, aim is not clear, and no hypothesis. The goal and hypothesis were clearly reported
Method:
All the required details are clearly given in the method section, while most of the information regarding the patients is given in the results section
Early in the text, highlight the important study design components.
Describe the context, the places, the pertinent times, such as the recruiting, exposure, follow-up, and data collection times. These data are given
All patients e.g. taken from pisan hospital. Who did diagnosis. Who collected information. Ethical issues with consent? Give the qualifying requirements, as well as the sources and procedures used to choose the participants. These data are reported.
Are people with comorbidities excluded e.g. anxiety, ocd, ed. These informatin is given in the Results section, togethet with their precsribed psychotropc drugs “Twenty-four (42.1%) patients were suffering from bipolar disorders (BDs) and 33 (57.9%) from a depressive disorder (MDD). Twenty-five (43.9%) patients were at their first major depressive episode, while the remaining 32 (56.1%) had a history of mood disorders (MDs). Fifty (87.7%) patients had no psychiatric comorbidity, while seven (12.3%) had at least one comorbidity, of whom, four (7.0%) with panic disorder, and three (5.3%) with generalized anxiety disorder. Thirty-five (61.4%) reported at least one medical comorbidity, and 22 patients (38.6%) had no medical disease. Cardiovascular disease was present in 27 (47.4%) and diabetes mellitus in four patients (7.0%). Five patients (8.8%) were also suffering from a neurological disease (stroke, essential tremor, Parkinson's disease, chorea) (Table 1). At the time of the observation, the patients were treated with different psychotropic drugs, according to the current episode and the possible longitudinal mental disorder. Thirty-five patients (61.4%) were treated with a mood stabilizer (valproic acid, lithium salts, gabapentin), and six of these (10.5%) with two mood-stabilizers. Forty-nine patients (86%) were treated with antidepressants (SSRIs, SNRIs, tricyclics, mirtazapine), 22 (38.6%) with an antipsychotic (quetiapine, clozapine, perphenazine, olanzapine, risperidone), 14 with benzodiazepines (24.6%), and one patient (1.8%) with an acetylcholinesterase inhibitor (AchEI).”
All outcomes, exposures, predictors, potential confounders, and effect modifiers should be precisely defined. Describe the diagnostic standards, if any. These details are reported.
Describe how the study size was determined. Describe the methods used to handle quantitative variables in the analyses. Describe the classifications that were chosen, if relevant, and why. All demographic and clinical data were presented for continuous variables in terms of mean ± standard deviation (SD), variation range (min and max values), or medians, when required. Comparisons for continuous variables were performed with the independent-sample Student’s t-test. The Kolmogorov-Smirnov test was used to determine the normality of distribution of the variables and therefore whether to use parametric or non-parametric tests.
Describe all statistical techniques, including confounding correction techniques, 2.3 Statistical analyses is data normal? If Yes, Why both mean and median used? We made the required corrections by including the software used to analyze the data (IBM SPSS, Version 27.0). The variables, depending on the type (categorical or continuous), were analyzed using the appropriate tests. Cohen's d standard was used to measure the effect size.
Describe any techniques used to study interactions and subgroups As explained in the statistical analyses section the comparisons between variables, since they do not have a normal distribution (Kolmogorov-Smirnov test), were carried out using the Wilcoxon/Mann-Whitney test (it is indicated in subparagraph 3.3).
Summarize the main findings in relation to the study's goals. We summarized the main findings
Describe the study's limitations while considering any possible bias or imprecision sources. Discuss any potential bias's magnitude and direction. Give a cautious overall interpretation of the findings while considering the goals, restrictions, variety of analyses, outcomes from related studies, and other pertinent data. Discuss how broadly (externally valid) the study's findings can be applied.The limitations are listedt in the text.,s follows “There are some limitations of this study that should be acknowledged, such as the small sample size, the use of largely self-administered tests, and the scanty use of instrumental investigations that might detect early cognitive ageing. Moreover, it is worth underlining that almost half of the subjects were suffering from BDs, while entailing the presence of possible cognitive alterations caused by previous manic episodes that were not analyzed, since we focused on the current depressive episode. Again, given the small sample size, we could not analyze the possible impact of different drugs or drug combinations”
We are very cautious in the interpretation of our findings that, nevertheless, remins interesting, as follows: “To sum up, our sample of elderly outpatients suffering from a moderate depression, mainly treated with mood stabilizers and antidepressants, showed mild alterations in cognitive functions that resulted only partially related the severity of the depressive episode. However, these dysfunctions became more evident when, besides cognitive functions, the behavioral responses were analyzed. A high educational qualification and having a partner seemed to represent protective factors against cognitive decline, while educational qualification did not. In addition, our data seem to suggest that previous episodes do not imply a more severe depression or a worse cognitive functioning. This might be explained by the protective and neurotrophic functions exerted by psychotropic drugs against brain ageing. However, their role in preventing the onset of dementia remain controversial. Although the early treatment of depressive symptoms seems to be useful not only for depression itself, but for the quality of life of the patients [altri già citati, e Voros et l, 2020], recent studies raise the question if antidepressants (at least those with anticholinergic activity) increase or not the risk of dementia in elderly depresseded subjects (Almeida, 2017, Kodesh; Solomonov)”.
2.2 Assessment scales pls provide better description of each tool and alpha for Italian scales. The scales have been widely described and we do not deem necessary to add too many details that are easily avialable in the specific references.
Results essential tremor need to be referred to as benign essential tremor
Please note that both expressions are correct.
Family history of psychiatric disorder? Who is family?-
Family history of psychiatric disorder was used in its most common and acknowledged notion that includes all the first grade relatives.
Table 3 pls convert to correlogram to avoid use of confusing panels. We prefer to omaintain table 3 as it is.
Add doi to references. According to published papers in the journal and the journal guidelines, DOI is not mandatory, so that we included it when available
Reviewer 4 Report
28 June 2023
Manuscript ID: brainsci-2490213
Type: Article
Title: “The complex relationship between depression and pseudodementia. A study on Italian outpatients” by Buccianelli B et al., submitted to Brain Sciences
Dear Authors,
Understanding and untangling the complex relationship between depression and pseudodementia, as well as identifying effective strategies for diagnosis, treatment, and management of these conditions, are the current challenges. In the present research article, entitled "The complex relationship between depression and pseudodementia: A study on Italian outpatients," Buccianelli and colleagues discuss the concept of "pseudodementia," a condition characterized by the coexistence of depressive symptoms and cognitive impairment that mimics dementia but is reversible. The study aimed to evaluate the presence of neurological alterations and cognitive impairment in elderly individuals experiencing depressive symptoms. Here, the authors used various assessment scales to measure depression severity, cognitive function, and other relevant factors. The results showed that pseudodementia can be distinguished from dementia based on clinical history, response to treatment, and neuroimaging findings. The study highlights the importance of differentiating between depression and dementia to provide appropriate treatment. The authors also discuss the potential influence of depression on cognitive decline and the need for early diagnosis to prevent dementia. Overall, the study provides insights into the complex relationship between depression and pseudodementia and suggests tailored therapeutic interventions to improve both conditions.
The strength of this manuscript lies in its comprehensive analysis of the complex interplay between depression and pseudodementia, drawing on a multidisciplinary approach and providing valuable insights into the diagnosis, treatment, and management of these conditions.
In general, I think the idea of this article is really interesting, and the authors’ fascinating observations on this timely topic may be of interest to the readers of Brain Sciences. However, some comments, as well as some crucial evidence that should be included to support the author’s argumentation, needed to be addressed to improve the quality of the manuscript, its adequacy, and its readability prior to publication in the present form, in particular by reshaping parts of the Introduction and Methods sections by adding more evidence and theoretical constructs.
Please consider the following comments:
1. I suggest changing the title. In my opinion, in the present form, it does seem too specific not to provide enough details about the relationship explored in the study. Are the Authors examining the correlation, causation, or potential mechanisms? Suggestions: "Shedding Light on the Enigmatic Connection: A Study on the Complex Relationship between Depression and Pseudodementia in Italian Outpatients"; "Depression and Pseudodementia: Decoding the Intricate Bonds in an Italian Outpatient Setting" [1-3].
2. AbstractAccording to the Journal’s guidelines, the abstract should be a total of about 200 words [4]. In my opinion, the authors should consider rephrasing this section. The abstract should contain most of the following kinds of information in brief form: Please consider giving a more synthetic overview of the paper's key points: I would suggest rephrasing the results and conclusion to make them easier for readers to understand. That being said, the authors should consider rephrasing this section. According to the journal’s guidelines, the abstract should contain most of the following kinds of information in brief form: Please consider giving a more synthetic overview of the paper's key points: I would suggest rephrasing the results and conclusion to make them easier for readers to understand. Overall, I would like the authors to present the background, the methods, the results, and the conclusion proportionally. The background should include the general background (one to two sentences), the specific background (two to three sentences), and the issue addressed by this review (one sentence) leading up to the objectives. In this section, I'd like the authors to provide background information, a problem statement, and an explanation of why they're breaking off. The results subsection ends with a sentence that puts this subsection in a general context. The conclusion should contain a single sentence describing the main result using language such as "Here we show." The conclusion should describe the study's potential and its contribution to the field, as well as provide a broader perspective (two to three sentences) that is understandable to a scientist from any discipline [5-7].
3. Keywords: Please list ten keywords chosen from Medical Subject Headings (MeSH) and major relevant indexes and use as many as possible in the title and in the first two sentences of the abstract [8,9].
4. A graphical abstract that will visually summarize the main findings of the manuscript is highly recommended.
5. In general, I recommend authors to use more references to back their claims, especially in the Introduction of this meta-analysis, which I believe is lacking. Thus, I recommend the authors to attempt to expand the topic of their article, as the bibliography is too concise. Nevertheless, I believe that less than 60/70 articles are too low for a research article. Therefore, I suggest the authors to focus their efforts on researching relevant literature: in my opinion, adding more citations will help to provide better and more accurate background to this study.
6. Introduction: I would like the authors to reorganize this section with about 1000 words and several paragraphs, introduce information on the key study constructs that readers in any discipline should understand, and make it persuasive enough to advance the main goal of the author's recent research and the specific goal the author has intended by this review. I'd like to suggest that the authors present the introduction beginning with the overall context, moving on to the specific context, and concluding with the current problem addressed in this study before moving on to the objectives. Those key structures ought to be set up logically and coherently [10].
7. In this regard, I recommend the authors reorganize the introduction section, which seems inhomogeneous and dispersive and, specifically, not as informative as an Introduction should be. The introduction could benefit from a discussion on the neural underpinnings of pseudodementia, as, in my opinion, it would be valuable to provide an overview of the brain regions and circuits that are typically affected in individuals with pseudodementia, such as the prefrontal cortex and the limbic system. Furthermore, to enhance the reader's understanding, I would suggest considering including a brief explanation of the neuropathological changes commonly associated with pseudodementia and describing the specific neurodegenerative processes, such as the accumulation of amyloid plaques or tau tangles, to provide a clearer context for the subsequent research (DOI: 10.3390/ijms24044114). Finally, consider incorporating a paragraph that highlights the significance of studying the neural mechanisms of pseudodementia and discussing how unraveling the neurobiological basis can lead to a better understanding of the condition, improved diagnostic accuracy, and the development of targeted interventions would enhance the overall impact of the research (https://doi.org/10.1016/j.neubiorev.2023.105163).
8. Materials and Methods: I recommend opening this section with a short introductory paragraph regarding the study design and citing more references to ensure the reliability and integrity of the evidence in the study design the authors built and the methodology they have decided to apply.
9. Patients: This section lacks detailed information about the sample selection process, inclusion and exclusion criteria, and potential biases. Additionally, the sample size is relatively small, which raises concerns about the generalizability of the findings.
10. Statistical analyses: The manuscript does not clearly describe the statistical analysis methods used to analyze the data. The authors should provide a detailed explanation of the statistical tests employed and justify their choices. Additionally, the results section should include appropriate statistical measures, such as effect sizes and confidence intervals.
11. Results: I suggest rewriting this section more accurately. To properly present experimental findings, I think that authors should provide full statistical details (like degree of freedom or post-hoc utilization) to ensure in-depth understanding and replicability of the findings. In addition, I would like the authors to close this section with a paragraph which puts the results into a more general context.
12. Discussion: I recommend that the authors reorganize this section with up to 1500 words, clarifying the following essential elements for discussion. Consider organizing this section into subsections based on the different topics or findings that should be addressed. Starting with an introductory paragraph, I would like the authors to present the summary of the previous section and to develop argument on the potential of this study complementing as the extension of the previous work, the implication of the findings of this study, how this study could facilitate future research, the ultimate goal, the challenge, the knowledge and the technology necessary to achieve this goal, the statement about this field in general, and finally the importance of this line of research [11,12].
13. In this regard, the authors discuss the need for a multidisciplinary approach in the diagnosis and treatment of elderly patients with pseudodementia. Still, I believe that it would be helpful to provide more specific recommendations for clinicians and highlight potential areas for future research. For example, I suggest discussing the implications of the findings on treatment approaches, therapeutic interventions, or the development of more accurate assessment tools would be valuable. Also, it would be beneficial to further explore the implications of the findings on treatment approaches from a neural perspective and discuss how a better understanding of the neural mechanisms involved in pseudodementia could inform the development of targeted interventions (DOI: 10.3390/biomedicines11030945). For instance, here the Authors could address the possibility of the investigation of the potential benefits of combining pharmacological treatments with cognitive training, neurostimulation techniques (like transcranial magnetic stimulation), or other interventions aimed at modulating neural activity and promoting cognitive recovery (https://doi.org/10.3389/fpsyt.2023.1225755).
14. In my opinion, the ‘Conclusions’ paragraph would benefit from a single paragraph that presents some thoughtful as well as in-depth considerations by the authors, because as it stands, it lists down all the main findings of the research without really stressing the theoretical significance of the study. Authors should make an effort to explain the theoretical implications as well as the translational application of their research.
15. References: The authors should consider revising the bibliography, as there are several incorrect citations. Indeed, according to the Journal’s guidelines, they should provide the abbreviated journal name in italics, the year of publication in bold, and the volume number in italics for all the references.
16. Finally, the manuscript does not clearly highlight the novelty or significance of the study. I would suggest the authors explicitly state the contribution of their research to the existing literature and explain how their findings advance the field.
Overall, the manuscript contains no figures, three tables, and 59 references. I believe that the manuscript may have important value in offering valuable insights and contributions to the field by highlighting the importance of promptly diagnosing and treating depressive symptoms in geriatric patients, emphasizing the need for a multidisciplinary approach, and advocating for the development of more accurate assessment tools and targeted pharmacological approaches. I hope that, after these careful revisions, this paper can meet the journal’s high standards for publication. I am available for a new round of revision of this paper.
I declare no conflict of interest regarding this manuscript.
Best regards,
Reviewer
References:
- https://plos.org/resource/how-to-write-a-great-title/
- https://www.nature.com/nature-index/news-blog/how-to-write-a-good-research-science-academic-paper-title
- https://www.indeed.com/career-advice/career-development/catchy-title
- https://www.mdpi.com/journal/brainsci/instructions
- https://www.scribbr.com/dissertation/abstract/
- https://writing.wisc.edu/handbook/assignments/writing-an-abstract-for-your-research-paper/
- https://pubmed.ncbi.nlm.nih.gov/30930712/
8. https://www.ncbi.nlm.nih.gov/pmc/articles/PMC7144240/
- https://meshb.nlm.nih.gov/
- https://dept.writing.wisc.edu/wac/writing-an-introduction-for-a-scientific-paper/
- https://doi.org/10.3163/1536-5050.103.2.001
- https://www.scribbr.com/dissertation/discussion/
28 June 2023
Manuscript ID: brainsci-2490213
Type: Article
Title: ‘The complex relationship between depression and pseudodementia. A study on Italian outpatients by Buccianelli B et al., submitted to Brain Sciences
Dear Authors,
After evaluating the English proficiency, it has been determined that some minor revisions to the English language are necessary. While the overall communication is clear and understandable, certain areas could benefit from slight improvements in grammar, syntax, and word choice. Paying attention to detail, such as refining sentence structure and ensuring proper tense usage, will enhance the coherence and fluency of the written work as a whole. Making minor editing adjustments can lead to an improvement in English language proficiency.
Best regards,
Reviewer
Author Response
Reviewer 4
We thank very much the reviewer 4 for stimulting us to improving our paper with important suggestions
- I suggest changing the title. In my opinion, in the present form, it does seem too specific not to provide enough details about the relationship explored in the study. Are the Authors examining the correlation, causation, or potential mechanisms? Suggestions: "Shedding Light on the Enigmatic Connection: A Study on the Complex Relationship between Depression and Pseudodementia in Italian Outpatients"; "Depression and Pseudodementia: Decoding the Intricate Bonds in an Italian Outpatient Setting" [1-3]. → We changed the title of the with the latter proposed by the reviewer that we mostly appreciated ("Depression and Pseudodementia: Decoding the Intricate Bonds in an Italian Outpatient Setting"), since we found it more pertinent and illustrative of our work.
- Abstract According to the Journal’s guidelines, the abstract should be a total of about 200 words [4]. In my opinion, the authors should consider rephrasing this section. The abstract should contain most of the following kinds of information in brief form: Please consider giving a more synthetic overview of the paper's key points: I would suggest rephrasing the results and conclusion to make them easier for readers to understand. That being said, the authors should consider rephrasing this section. According to the journal’s guidelines, the abstract should contain most of the following kinds of information in brief form: Please consider giving a more synthetic overview of the paper's key points: I would suggest rephrasing the results and conclusion to make them easier for readers to understand. Overall, I would like the authors to present the background, the methods, the results, and the conclusion proportionally. The background should include the general background (one to two sentences), the specific background (two to three sentences), and the issue addressed by this review (one sentence) leading up to the objectives. In this section, I'd like the authors to provide background information, a problem statement, and an explanation of why they're breaking off. The results subsection ends with a sentence that puts this subsection in a general context. The conclusion should contain a single sentence describing the main result using language such as "Here we show." The conclusion should describe the study's potential and its contribution to the field, as well as provide a broader perspective (two to three sentences) that is understandable to a scientist from any discipline [5-7]. The abstract was rephrase and better organized
- Keywords: Please list ten keywords chosen from Medical Subject Headings (MeSH) and major relevant indexes and use as many as possible in the title and in the first two sentences of the abstract [8,9].The key words are listed and used throughout the text
- A graphical abstract that will visually summarize the main findings of the manuscript is highly recommended.
- In general, I recommend authors to use more references to back their claims, especially in the Introduction of this meta-analysis, which I believe is lacking. Thus, I recommend the authors to attempt to expand the topic of their article, as the bibliography is too concise. Nevertheless, I believe that less than 60/70 articles are too low for a research article. Therefore, I suggest the authors to focus their efforts on researching relevant literature: in my opinion, adding more citations will help to provide better and more accurate background to this study.
More refererences have been added throughout the paper. Please note, our paper is not a meta-analysis.
- Introduction:
We have carefully revised and reorganized the introduction as suggested to make it more consistent,coherent and focused
I would like the authors to reorganize this section with about 1000 words and several paragraphs, introduce information on the key study constructs that readers in any discipline should understand, and make it persuasive enough to advance the main goal of the author's recent research and the specific goal the author has intended by this review. I'd like to suggest that the authors present the introduction beginning with the overall context, moving on to the specific context, and concluding with the current problem addressed in this study before moving on to the objectives. Those key structures ought to be set up logically and coherently [10].
- In this regard, I recommend the authors reorganize the introduction section, which seems inhomogeneous and dispersive and, specifically, not as informative as an Introduction should be. The introduction could benefit from a discussion on the neural underpinnings of pseudodementia, as, in my opinion, it would be valuable to provide an overview of the brain regions and circuits that are typically affected in individuals with pseudodementia, such as the prefrontal cortex and the limbic system. Furthermore, to enhance the reader's understanding, I would suggest considering including a brief explanation of the neuropathological changes commonly associated with pseudodementia and describing the specific neurodegenerative processes, such as the accumulation of amyloid plaques or tau tangles, to provide a clearer context for the subsequent research (DOI: 10.3390/ijms24044114). This is out of the scope of our paper nd tehreference indicated is not congruent as it is entitled “Exploring Novel Therapeutic Targets in the Common Pathogenic Factors in Migraine and Neuropathic Pain”
Finally, consider incorporating a paragraph that highlights the significance of studying the neural mechanisms of pseudodementia and discussing how unraveling the neurobiological basis can lead to a better understanding of the condition, improved diagnostic accuracy, and the development of targeted interventions would enhance the overall impact of the research (https://doi.org/10.1016/j.neubiorev.2023.105163). The main problem is to study the neurobiology of dementia AND of depression as pseudodemnti is a controvesial topic.
- Materials and Methods: I recommend opening this section with a short introductory paragraph regarding the study design and citing more references to ensure the reliability and integrity of the evidence in the study design the authors built and the methodology they have decided to apply.
The study design and scopes are present the end of the Introduction.
- Patients: This section lacks detailed information about the sample selection process, inclusion and exclusion criteria, and potential biases. All the informationrequired is now present either in the methods or in the results section
Additionally, the sample size is relatively small, which raises concerns about the generalizability of the findings. The sample size is not large, but sufficient to perform reliable statistical analyses.
- Statistical analyses: The manuscript does not clearly describe the statistical analysis methods used to analyze the data. The authors should provide a detailed explanation of the statistical tests employed and justify their choices. Additionally, the results section should include appropriate statistical measures, such as effect sizes and confidence intervals.
We have made the required corrections by including the software used to analyze the data (IBM SPSS, Version 27.0). The variables, depending on the type (categorical or continuous), were analyzed using the appropriate tests; moreover, by applying the Kolmogorov-Smirnov test, we established when to use non-parametric or parametric tests. Cohen's d standard was used to measure the effect size.
- Results: I suggest rewriting this section more accurately. To properly present experimental findings, I think that authors should provide full statistical details (like degree of freedom or post-hoc utilization) to ensure in-depth understanding and replicability of the findings. In addition, I would like the authors to close this section with a paragraph which puts the results into a more general context.
The methods used for data analysis are explained individually in the paragraph "statistical analysis" while in subparagraph 3.3 the method used to calculate the correlations between groups is described. Finally, the results are summarized in the "discussion". We think that creating a new subsection that again summarizes all the results could be redundant in writing the data.
- Discussion: This section has been corrected and modfied to make it more logical and consequential.
I recommend that the authors reorganize this section with up to 1500 words, clarifying the following essential elements for discussion. Consider organizing this section into subsections based on the different topics or findings that should be addressed. Starting with an introductory paragraph, I would like the authors to present the summary of the previous section and to develop argument on the potential of this study complementing as the extension of the previous work, the implication of the findings of this study, how this study could facilitate future research, the ultimate goal, the challenge, the knowledge and the technology necessary to achieve this goal, the statement about this field in general, and finally the importance of this line of research [11,12].
- In this regard, the authors discuss the need for a multidisciplinary approach in the diagnosis and treatment of elderly patients with pseudodementia. Still, I believe that it would be helpful to provide more specific recommendations for clinicians and highlight potential areas for future research. For example, I suggest discussing the implications of the findings on treatment approaches, therapeutic interventions, or the development of more accurate assessment tools would be valuable. Also, it would be beneficial to further explore the implications of the findings on treatment approaches from a neural perspective and discuss how a better understanding of the neural mechanisms involved in pseudodementia could inform the development of targeted interventions (DOI: 10.3390/biomedicines11030945). For instance, here the Authors could address the possibility of the investigation of the potential benefits of combining pharmacological treatments with cognitive training, neurostimulation techniques (like transcranial magnetic stimulation), or other interventions aimed at modulating neural activity and promoting cognitive recovery (https://doi.org/10.3389/fpsyt.2023.1225755).
- In my opinion, the ‘Conclusions’ paragraph would benefit from a single paragraph that presents some thoughtful as well as in-depth considerations by the authors, because as it stands, it lists down all the main findings of the research without really stressing the theoretical significance of the study. Authors should make an effort to explain the theoretical implications as well as the translational application of their research. The conlcuiosn have been significantly modified to accmpish gthe reviewee’s request
- References: The authors should consider revising the bibliography, as there are several incorrect citations. Indeed, according to the Journal’s guidelines, they should provide the abbreviated journal name in italics, the year of publication in bold, and the volume number in italics for all the references
References were adjusted according to the journal’s guidelines.
- Finally, the manuscript does not clearly highlight the novelty or significance of the study. I would suggest the authors explicitly state the contribution of their research to the existing literature and explain how their findings advance the field. This has been clearly highlighted in the revised paper
Overall, the manuscript contains no figures, three tables, and 59 references. I believe that the manuscript may have important value in offering valuable insights and contributions to the field by highlighting the importance of promptly diagnosing and treating depressive symptoms in geriatric patients, emphasizing the need for a multidisciplinary approach, and advocating for the development of more accurate assessment tools and targeted pharmacological approaches. I hope that, after these careful revisions, this paper can meet the journal’s high standards for publication. I am available for a new round of revision of this paper. As mentioned above, more refererences have been added
Round 2
Reviewer 3 Report
thanks, no further comments
Author Response
There is no questions from the reviewer 3
Reviewer 4 Report
2 August 2023
Manuscript ID: brainsci-2490213
Type: Article
Title: “The complex relationship between depression and pseudodementia. A study on Italian outpatients” by Buccianelli B et al., submitted to Brain Sciences
Dear Authors,
In the present research article, entitled "The complex relationship between depression and pseudodementia: A study on Italian outpatients," Buccianelli and colleagues evaluate the presence of neurological alterations and cognitive impairment in elderly individuals experiencing depressive symptoms. I am pleased to see that the authors have attempted to revise the manuscript in the peer review session. Nevertheless, the revisions remain partial in regard to my previous report. Prior to publication, I respectfully request that the authors consider my comments and revise the manuscript to meet the high standards of the journal. In addition, I anticipate the authors preparing “a detailed point-wise rebuttal” to my remarks Comments:
1. Abstract: As I suggested in the previous round, I would like the authors to pay special attention to the following crucial elements for this section: According to the Journal’s guidelines, the abstract should be a total of about 200 words [1]. I would like the authors to present the background, the methods, the results, and the conclusion proportionally. The background should include the general background (one to two sentences), the specific background (two to three sentences), and the issue addressed by this review (one sentence) leading up to the objectives. In this section, I'd like the authors to provide background information, a problem statement, and an explanation of why they're breaking off. The results subsection ends with a sentence that puts this subsection in a general context. The conclusion should contain a single sentence describing the main result using language such as "Here we show." The conclusion should describe the study's potential and its contribution to the field, as well as provide a broader perspective (two to three sentences) that is understandable to a scientist from any discipline [2–4].
2. Keywords: Please list ten keywords chosen from Medical Subject Headings (MeSH) and major relevant indexes and use as many as possible in the title and in the first two sentences of the abstract [5,6].
3. A graphical abstract that will visually summarize the main findings of the manuscript is highly recommended. I did not receive any response to this point.
4. Introduction: I would like the authors to reorganize this section and clarify the objectives in the end of this section: I would like the authors to reorganize this section with about 1000 words and several paragraphs, introduce information on the key study constructs that readers in any discipline should understand, and make it persuasive enough to advance the main goal of the author's recent research and the specific goal the author has intended by this review. I'd like to suggest that the authors present the introduction beginning with the overall context, moving on to the specific context, and concluding with the current problem addressed in this study before moving on to the objectives. Those key structures ought to be set up logically and coherently [7].
5. Materials and Methods: I recommend opening this section with a short introductory paragraph regarding the study design before describing the details of methodology.
6. Results: I suggest presenting figures summarizing the results and I would like the authors to close this section with a paragraph which puts the results into a more general context.
7. Discussion: I would like the authors to reorganize this section by paying special attention to the main structure of this section as suggested previously: I recommend that the authors reorganize this section with up to 1500 words, clarifying the following essential elements for discussion. Consider organizing this section into subsections based on the different topics or findings that should be addressed. Starting with an introductory paragraph, I would like the authors to present “the summary of the previous section” and to develop argument on the potential of this study complementing as the extension of the previous work, the implication of the findings of this study, how this study could facilitate future research, the ultimate goal, the challenge, the knowledge and the technology necessary to achieve this goal, the statement about this field in general, and finally the importance of this line of research [8,9].
8. Please clarify this point: Finally, the manuscript does not clearly highlight the novelty or significance of the study. I would suggest the authors explicitly state the contribution of their research to the existing literature and explain how their findings advance the field.
Overall, the manuscript contains no figures, three tables, and 79 references. I believe that the manuscript may have important value in offering valuable insights and contributions to the field by highlighting the importance of promptly diagnosing and treating depressive symptoms in geriatric patients, emphasizing the need for a multidisciplinary approach, and advocating for the development of more accurate assessment tools and targeted pharmacological approaches. I hope that, after these careful revisions, this paper can meet the journal’s high standards for publication. I am available for a new round of revision of this paper.
I declare no conflict of interest regarding this manuscript.
Best regards,
Reviewer
References:
- https://www.mdpi.com/journal/brainsci/instructions
- https://www.scribbr.com/dissertation/abstract/
- https://writing.wisc.edu/handbook/assignments/writing-an-abstract-for-your-research-paper/
- https://doi.org/10.4103/sja.SJA_685_18
- https://meshb.nlm.nih.gov/
- https://doi.org/10.5812/ijem.100159
- https://dept.writing.wisc.edu/wac/writing-an-introduction-for-a-scientific-paper/
- https://doi.org/10.3163/1536-5050.103.2.001
- https://www.scribbr.com/dissertation/discussion/
2 August 2023
Manuscript ID: brainsci-2490213
Type: Article
Title: ‘The complex relationship between depression and pseudodementia. A study on Italian outpatients by Buccianelli B et al., submitted to Brain Sciences
Dear Authors,
On the basis of the English proficiency evaluation, it has been determined that some minor editing of the English language is necessary. While the overall communication is clear and comprehensible, some areas could benefit from minor improvements in grammar, syntax, and word selection. Attention to detail, such as refining sentence structure and ensuring correct tense usage, will improve the coherence and fluency of the written work as a whole. The English language proficiency can be improved with some minor editing adjustments.
Best regards,
Reviewer
Author Response
Answers to specific questions
- Abstract: As I suggested in the previous round, I would like the authors to pay special attention to the following crucial elements for this section: According to the Journal’s guidelines, the abstract should be a total of about 200 words [1]. I would like the authors to present the background, the methods, the results, and the conclusion proportionally. The background should include the general background (one to two sentences), the specific background (two to three sentences), and the issue addressed by this review (one sentence) leading up to the objectives. In this section, I'd like the authors to provide background information, a problem statement, and an explanation of why they're breaking off. The results subsection ends with a sentence that puts this subsection in a general context. The conclusion should contain a single sentence describing the main result using language such as "Here we show." The conclusion should describe the study's potential and its contribution to the field, as well as provide a broader perspective (two to three sentences) that is understandable to a scientist from any discipline [2–4]. We modified the abstract and tried to accomplish the reviewer’s requests, according to the limitation of about 200 words (our abstract contains 216 words)
- Keywords: Please list ten keywords chosen from Medical Subject Headings (MeSH) and major relevant indexes and use as many as possible in the title and in the first two sentences of the abstract [5,6]. The keywords have been updated according to the MESH
- A graphical abstractthat will visually summarize the main findings of the manuscript is highly recommended. I did not receive any response to this point. The graphical abstract has been done and inserted in the first page.
- Introduction: I would like the authors to reorganize this section and clarify the objectives in the end of this section: I would like the authors to reorganize this section with about 1000 words and several paragraphs, introduce information on the key study constructs that readers in any discipline should understand, and make it persuasive enough to advance the main goal of the author's recent research and the specific goal the author has intended by this review. I'd like to suggest that the authors present the introduction beginning with the overall context, moving on to the specific context, and concluding with the current problem addressed in this study before moving on to the objectives. Those key structures ought to be set up logically and coherently [7]. This section has been significantly revised and lengthened, with a more logical sequence.
- Materials and Methods: I recommend opening this section with a short introductory paragraph regarding the study design before describing the details of methodology. The short introductory paragraph has been added.
- Results: I suggest presenting figures summarizing the results and I would like the authors to close this section with a paragraph which puts the results into a more general context. Two figures have been added. .Although, in our opinion, it is not customary to comment on the overall findings in the results section., w eadded ageneral summarizing sentence.
- Discussion: I would like the authors to reorganize this section by paying special attention to the main structure of this section as suggested previously: I recommend that the authors reorganize this section with up to 1500 words, clarifying the following essential elements for discussion. Consider organizing this section into subsections based on the different topics or findings that should be addressed. Starting with an introductory paragraph, I would like the authors to present “the summary of the previous section” and to develop argument on the potential of this study complementing as the extension of the previous work, the implication of the findings of this study, how this study could facilitate future research, the ultimate goal, the challenge, the knowledge and the technology necessary to achieve this goal, the statement about this field in general, and finally the importance of this line of research [8,9]. We significantly revised the Discussion, while taking care of the suggestions of the revewier. Specifically, after the introductory remarks, we ccommented on our finding along their sequence reported in the Results section.
- Please clarify this point: Finally, the manuscript does not clearly highlight the novelty or significance of the study. I would suggest the authors explicitly state the contribution of their research to the existing literature and explain how their findings advance the field. We highlighted that this study is the first one carried out in our country in a sample of depressed patients
Overall, the manuscript contains no figures, three tables, and 79 references. I believe that the manuscript may have important value in offering valuable insights and contributions to the field by highlighting the importance of promptly diagnosing and treating depressive symptoms in geriatric patients, emphasizing the need for a multidisciplinary approach, and advocating for the development of more accurate assessment tools and targeted pharmacological approaches. I hope that, after these careful revisions, this paper can meet the journal’s high standards for publication. I am available for a new round of revision of this paper. Some figures have been added to make the results more visible
Other references have been added in the Introduction and Discussion (88 in total)
